

# A study of measurement scenarios for the future CO2M mission: avoidance of detector saturation and the impact on XCO2 retrievals

Michael Weimer[1], Michael Hilker[1], Stefan Noël[1], Max Reuter[1], Michael Buchwitz[1], Blanca Fuentes Andrade[1], Rüdiger Lang[2], Bernd Sierk[2], Yasjka Meijer[3], Heinrich Bovensmann[1], John P. Burrows[1], and Hartmut Bösch[1]

[1]Institute of Environmental Physics, University of Bremen, Bibliothekstraße 1, 28359 Bremen, Germany
[2]EUMETSAT, EUMETSAT-Allee 1, 64295 Darmstadt, Germany
[3]European Space Agency, Keplerlaan 1, 2201 AZ Noordwijk, the Netherlands

**Correspondence:** Michael Weimer (mweimer@iup.physik.uni-bremen.de)

**Abstract.** The human direct and indirect release of carbon dioxide ($CO_2$) into the atmosphere is the main driver of the anthropogenic change in climate since the industrial revolution. The Paris agreement from 2015 requires regular country-based reports of greenhouse gas emissions. Inverse modeling of observed concentrations of greenhouse gases is one important approach to verify the reported emissions. The future constellation of Copernicus Anthropogenic $CO_2$ Monitoring (CO2M) satellites is dedicated to greenhouse gas measurements with high spectral and spatial resolution and coverage. The requirements for the performance of the instruments and retrieval algorithms for the column-averaged dry-air mole fraction ($XCO_2$) are stringent in order to identify, assess and monitor the $CO_2$ emissions from space. In this study, we analyze the impact of avoiding detector saturation on the precision and sampling of $XCO_2$. We use the Fast atmOspheric traCe gAs retrievaL (FOCAL) algorithm which has been selected to be one of the operational greenhouse gas retrieval algorithms to be implemented within the CO2M ground segment. In order to avoid saturation, the number of read-outs per sampling time can be increased and the signals can be co-added onboard, which we refer to as "temporal oversampling" in this study. We use a subsampled one-year dataset of simulated radiances to define the temporal oversampling factors (OSFs) that are sufficient to avoid detector sarutarion and then apply the defined OSF combinations globally. We find that OSFs larger than one will lead to a significant decrease in number of saturated observations with some impact on the median $XCO_2$ precision, concluding that OSFs larger than one should be considered for the satellite mission. These results are based on simulated radiances. Consequently, the real impact on the precision should be analyzed in more detail during the commissioning phase of the satellite.

## 1 Introduction

It is now well established that the direct and indirect human release of carbon dioxide ($CO_2$), since the industrial revolution, is the most important cause of the recent climate change (IPCC, 2023). Due to its long projected and irreversible impact on global warming on a timescale of a millennium (e.g., Archer et al., 2009; Solomon et al., 2009) and its sources from fossil fuel combustion among others, reduction of $CO_2$ emissions is an internationally agreed environmental policy goal, as stated e.g. in the Paris Agreement from 2015 (UNFCCC, 2015). This agreement requires that countries report their emissions on a regular




basis. Atmospheric measurements of the $CO_2$ concentrations, including e.g. in-situ surface observations and satellite-based remote sensing instruments, combined with inverse modeling to determine surface fluxes offer a unique opportunity to verify 25 and support these reported emissions.

Space-borne total column $CO_2$ measurements have a long history starting with those retrieved from the pioneering instrument SCanning Imaging Absorption spectroMeter for Atmospheric CHartographY (SCIAMACHY, Burrows et al., 1995; Bovensmann et al., 1999; Buchwitz et al., 2005; Reuter et al., 2010; Schneising et al., 2011) and other satellites such as the Greenhouse gases Observing SATellite (GOSAT) and GOSAT-2 (Kuze et al., 2009; Nakajima et al., 2012), the Orbiting Carbon 30 Observatory (OCO) version 2 and 3 (Crisp, 2015; Taylor et al., 2020) and TanSat (Liu et al., 2018).

The Copernicus Anthropogenic $CO_2$ Monitoring (CO2M) mission is a future constellation of three identical satellites in a near-polar sun-synchronous orbit with an equator crossing time at 11:30 LT in a descending node (Janssens-Maenhout et al., 2020; Sierk et al., 2021; Meijer et al., 2023). The first satellite is planned to be launched in 2026. The mission builds on the concepts of CarbonSat with extended instrumentation (Bovensmann et al., 2010; Velazco et al., 2011; Buchwitz et al., 2013; 35 Broquet et al., 2018). Its primary instrument is a push-broom imaging spectrometer (CO2I) measuring solar radiances reflected at the Earth's surface and scattered in the atmosphere in three spectral bands: a) the near infrared (NIR, $747 - 773\,\text{nm}$), used to retrieve information about scattering properties or the atmospheric dry-air column density, aerosols and solar-induced fluorescence (SIF); b) and two bands in the short-wave infrared (SWIR1, $1590 - 1675\,\text{nm}$ and SWIR2, $1990 - 2095\,\text{nm}$), used to derive information about atmospheric $CO_2$, $CH_4$, aerosols and water vapor. As a result the satellites will enable the 40 determination of the column-averaged dry-air mole fraction of atmospheric $CO_2$ and $CH_4$, called $XCO_2$ and $XCH_4$ hereafter, at a total spatial sample size of about $4\,\text{km}^2$ and a swath width of around $250\,\text{km}$. This resolution and swath width is a trade-off between detection of local sources and a frequent global coverage, with some limitations, e.g. due to clouds covering the tropospheric signal. In addition to CO2I, the CO2M mission will enable the measurement of the $NO_2$ content in the atmosphere with a spectrometer in the visible spectral range (NO2I) and information about clouds in the atmosphere with a Cloud Imager 45 (CLIM) and about aerosols with a Multi-Angle Polarimeter (MAP), see also Meijer et al. (2023) for an overview.

The potential for $CO_2$ emission verification with CO2M has been shown by studies using simulated radiances (e.g., Kuhlmann et al., 2020) and using measurements of satellites already in operation (e.g., Reuter et al., 2019; Fuentes Andrade et al., 2024). Three retrieval algorithms are considered for the operational greenhouse gas product of CO2M with differences especially in the treatment of light scattering in the retrievals: the Remote sensing of Trace gas and Aerosol Product (RemoTAP, Lu 50 et al., 2022), the Flexible and Unified Spectral InversiON ALgorithm Platform (Fusional-P-UOL-FP) based on the algorithm described in Cogan et al. (2012) and the Fast atmOspheric traCe gAs retrievaL (FOCAL, Reuter et al., 2017a, b; Noël et al., 2021, 2022, 2024).

Quantifying anthropogenic $CO_2$ emissions from space is challenging because atmospheric signals resulting from these emissions are usually less than $1\,\%$ larger than the background (global $XCO_{2,\text{bg}} \approx 419\,\text{ppm}$ in 2023, Copernicus Climate 55 Change Service, 2024). In addition, the natural variability during the year is of similar order of magnitude (e.g., Forkel et al., 2016). Therefore, the precision ($0.7\,\text{ppm}$ for CO2I) and accuracy requirements ($0.5\,\text{ppm}$ for CO2I) to the instrument calibration and retrieval algorithms are stringent (ESA, 2020). Consequently, all aspects influencing the precision of the retrieved $XCO_2$





have to be considered and analyzed carefully in order to meet these requirements. Precision values between 0.4 and $0.6\,\mathrm{ppm}$ could be inferred from several retrieval algorithms for CO2M using one read-out per integration time step (Lu et al., 2022; Reuter et al., 2024). Noël et al. (2024) showed first performance assessments of the FOCAL version for CO2M with simulated radiances. Here, we consider the effect of detector saturation and investigate the impact of reducing the detector exposure time in the nadir configuration where the instrument's zenith angle is close to zero.

Detector saturation occurs when the number of photons collected by the detector is larger than the characteristic full well capacity (FWC), e.g. due to a bright surface like a desert. Saturation of the detectors results in several negative impacts, which lead to errors in the $CO_2$ retrieval. At signal levels above the FWC the detector typically exhibits strong non-linearity with quickly fading response towards saturation (e.g., Staebell et al., 2021, for the airborne instrument MethaneAIR). Consequently, saturation-affected measurements have to be removed (Yoshida et al., 2011; Kataoka et al., 2017; Tian et al., 2018; Shi et al., 2021) and should generally be avoided. The GOSAT instrument has different gain modes to avoid detector saturation (Kataoka et al., 2017; Reuter et al., 2012; Taylor et al., 2022). In glint geometry over ocean, where the satellite's field of view is shifted towards the sun-glint spot, it has been found that saturation can affect the measurements and can be avoided by looking near the glint spot but excluding it (Boesch et al., 2011; Eldering et al., 2012; Crisp et al., 2017). Saturation in general can also be avoided by reducing the exposure time of the detector, thereby increasing the maximum detectable radiance in that spectral band, but also impacting the retrieved $XCO_2$ precision (Nakajima et al., 2015; Grossmann et al., 2018; Staebell et al., 2021; Clavier et al., 2024). This is further discussed in Sect. 2.

The goal of this study is to define scenarios reducing the detector exposure time while maximizing the coverage and minimizing the negative impact of saturation on the $XCO_2$ precision. For this, we use simulated radiances calculated at the CO2M spatial samples with added noise corresponding to the respective detector setting. After defining detector saturation and its relation to the reduction of the exposure time in Sect. 2, we describe the simulated radiances used in this study (Sect. 3). As a next step, we define scenarios to determine the detector exposure time needed in each spectral band (see Sect. 4), then apply these scenarios to simulated radiances and retrieve $XCO_2$ using the FOCAL algorithm (Sect. 5). The impact of the scenarios on the global $XCO_2$ precision is discussed in Sect. 6. Section 7 provides some concluding remarks.

## 2 Detector design, saturation, oversampling factor, signal-to-noise ratio

The design of the CO2I/NO2I is comprised of four grating spectrometers sharing a common telescope, entrance slit and collimator, as described in Sierk et al. (2021). Here we briefly summarize the features that are relevant for the present study. The multi-band spectrometer operates according to the push-broom imaging principle: The entrance slit is projected onto the Earth's surface, defining the swath width in the across-track (ACT) direction. The CO2I design features a slit composed of a number of rectangular optical fibers, which are aligned to form an array of apertures defining the spatial samples. The fiber core dimensions define the spatial sample size in ACT direction ($326\,\mathrm{\mu m}$ corresponding to $1.8\,\mathrm{km}$ on Earth) and the instantaneous field-of-view (IFOV) in the along-track (ALT) direction ($124\,\mathrm{\mu m}$ corresponding to $814\,\mathrm{m}$ on Earth). The spatial sampling in



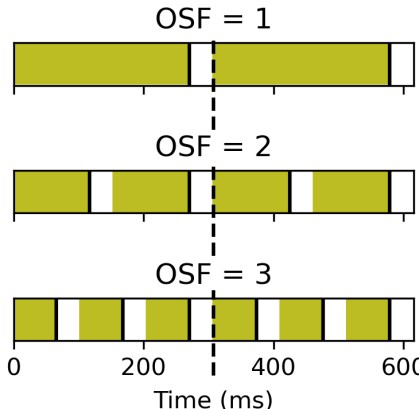

**Figure 1.** Illustration of the integration times (colored) for the oversampling factors that are foreseen for the detectors in the NIR and SWIR bands of CO2M. White spaces are times for reading out the signal (about $37\,\mathrm{ms}$). The end of the integration time is denoted by solid black lines at the end of each rectangle. Black dashed line: Sampling period $t_{\mathrm{samp}}$ of $308\,\mathrm{ms}$, see Eq. (2).

ALT direction is performed by the motion of the IFOV during the integration period $t_{\mathrm{int}}$, in which signal is accumulated on the detectors.

During the sampling time $t_{\mathrm{samp}} = 308\,\mathrm{ms}$ the slit projection on Earth moves by $2.2\,\mathrm{km}$, which defines the ALT ground sampling distance. A spatial sample therefore has the extent of about $1.8 \times 2.2 \approx 4\,\mathrm{km}^2$ (with small variation across the swath due to projection on the Earth's surface). At any instant in time, the detector pixels sample the image spatially in ACT direction 95 and spectrally in the perpendicular direction. From the signal of the dispersed light (in electrons), integrated during the sampling period, radiance spectra are derived, one for each fiber comprising the entrance slit.

The number of electrons accumulated by a detector pixel sampling the wavelength $\lambda$, denoted as signal $S(\lambda)$, depends on the ground scene as well as the properties of the spectrometer, and can be as expressed as:

$$S(\lambda) = L(\lambda) \cdot \eta \cdot \tau \cdot \Delta\lambda \cdot QE(\lambda) \cdot t_{\mathrm{int}} \tag{1}$$

Here, $L(\lambda)$ is the top-of-atmosphere spectral radiance, $\eta$ the étendue of the instrument (product of entrance pupil area and observation solid angle), $\tau$ the transmission of the optics, $\Delta\lambda$ the spectral bandwidth (or sampling interval) of the pixel, and $QE$ the quantum efficiency of the detector. $t_{\mathrm{int}}$ is the time during which light is accumulated within the sampling period. For CO2I, the sampling period ($t_{\mathrm{samp}}$) is $308\,\mathrm{ms}$. The signal integration for all CO2I detectors is paused during the read-out process. Accordingly, $t_{\mathrm{int}}$ as such is reduced by multiples of the read-out time $t_{\mathrm{RO}}$ of about $37\,\mathrm{ms}$:

$$t_{\mathrm{int}} = t_{\mathrm{samp}} - \mathrm{OSF} \cdot t_{\mathrm{RO}} \tag{2}$$

The factor OSF denotes the temporal oversampling factor, which is the number of detector read-outs within the sampling time. An illustration of the integration times for different OSFs can be found in Fig. 1. Multiple read-outs (OSF > 1) become necessary when the number of electrons accumulated during integration time exceeds the FWC of the detector pixels (called



saturation hereafter). Note that the gaps in signal integration indicated by the white spaces in Fig. 1 are short with respect to the
sampling time. They do not result in spatial under-sampling as the IFOV in ALT direction (approx. $800\,\mathrm{m}$) is larger than the
on-ground motion of the entrance slit image during the read-out time, which is about $200\,\mathrm{m}$. An OSF larger than one increases
the maximum radiance that can be measured by the detector, which is proportional to the time of each individual integration
time (colored in Fig. 1).

The detectors used for the two SWIR spectrometers of CO2I (Lynred NGP) feature a FWC of approximately 650.000
electrons. If the signal acquired during integration time exceeds this limit, the respective detector pixel becomes saturated.
Such pixels do not yield meaningful measurements, and a radiance spectrum with saturated pixels has to be discarded. In
order to avoid data loss, the detector can however be operated with OSF > 1, meaning that more than one read-out cycle
is performed within the sampling period of $t_{\mathrm{samp}} = 308\,\mathrm{ms}$ (Fig. 1). Apart from data loss over bright ground scenes, the
necessity for saturation avoidance by temporal oversampling (OSF > 1) arises from the inherent effect of instrument stray
light. Efficient correction of stray light from imperfect optical components (surface roughness and contamination), as well as
parasitic reflections between them (ghosts) is mandatory to achieve compliance with the stringent radiometric requirements
of the CO2M mission. Stray light correction algorithms require accurate knowledge of the signal distribution across the focal
plane, which is derived from the measured signal image. In the presence of signal saturation, and hence invalid radiation
measurements, such correction becomes inaccurate, if not infeasible, since the largest stray light contribution from the brightest
signals cannot be reconstructed. For the reasons outlined above, the CO2I spectrometer is likely to be operated with temporal
oversampling, leading to signal loss according to Eq. (2). In this study, we neglect the effect of saturation on neighbored spatial
samples and remove spatial samples in case of saturation in their spectrum.

A major drawback of oversampling is the loss of radiometric signal from the total read-out time $\mathrm{OSF} \cdot t_{\mathrm{RO}}$, which decreases
the signal-to-noise ratio (SNR) because the read-out noise increases with mutliple read-outs. In order to quantify the impact
of oversampling on SNR we further develop Eq. (1) to obtain an expression for the signal-to-noise ratio of the measured
radiances: The shot noise of the measurement, given by the square root of the signal $S(\lambda)$ combines with the signal-independent
components of the detection as

$$N_{\mathrm{total}} = \sqrt{S(\lambda) + (I_{\mathrm{dark}} + I_{\mathrm{Tb}}) \cdot t_{\mathrm{int}} + (N_{\mathrm{RO}}^2 + N_{\mathrm{AD}}^2 + N_{\mathrm{VC}}^2) \cdot \mathrm{OSF}}, \tag{3}$$

where $I_{\mathrm{dark}}$ is the dark current of the detector, $I_{\mathrm{Tb}}$ the shot noise from background thermal emission, $N_{\mathrm{RO}}$ its read-out
noise, $N_{\mathrm{AD}}$ the digitization noise, and $N_{\mathrm{VC}}$ the video chain noise. The SNR can then be expressed as

$$\mathrm{SNR} = \frac{A \cdot L}{\sqrt{A \cdot L + B}}, \tag{4}$$

in which the contributing noise sources are grouped into components scaling with the radiance $L(\lambda)$ and read as

$$A = \eta \cdot \tau \cdot \Delta\lambda \cdot QE(\lambda) \cdot [t_{\mathrm{samp}} - (\mathrm{OSF} \cdot t_{\mathrm{RO}})], \tag{5}$$

and signal-independent parameters determined by the detector and read-out-electronics

$$B = (I_{\mathrm{dark}} + I_{\mathrm{Tb}}) \cdot [t_{\mathrm{samp}} - (\mathrm{OSF} \cdot t_{\mathrm{RO}})] + (N_{\mathrm{RO}}^2 + N_{\mathrm{AD}}^2 + N_{\mathrm{VC}}^2) \cdot \mathrm{OSF}. \tag{6}$$



As can be seen, both the nominator and denominator in Eq. (4) are affected when the oversampling factor is increased: The signal is decreased by the multiple read-out times, in which no signal electrons are integrated, and the total noise is increased, as more read-out noise is accumulated. Both effects reduce the SNR of the measured radiance for OSF > 1.

In order to apply the different OSF scenarios to the radiances, we use $A$ and $B$ parameters provided to us by ESA (ESA, private communication, 2023) to compute the SNR in the data using Eq. (4). In the files, these parameters are given for an edge spatial sample and a center spatial sample at discrete integer wavelengths so that they have to be interpolated to all detector pixels. We used linear interpolation in both wavelength and across-track dimensions. The $A$ and $B$ parameters depend on the number of read-outs and thus on the used OSF so that the SNR is OSF-specific at each wavelength.

Optimization of in-flight operation calls for avoidance of saturation on the one hand, while maintaining the largest possible SNR of the measured radiance spectra on the other. This optimization requires a careful analysis of the expected radiance levels and their variation, based on the realistic simulation of ground scenes, which is the topic of this study.

## 3 Radiance spectra simulated with SCIATRAN

We base our investigations on simulated radiances at the CO2M spatial samples, which we use as input for the retrievals. With simulated radiances, we have exact control over the noise that is added to the radiances so that the impact of increasing the OSF can be separated from other instrumental effects. The same one-year subset radiances, simulated with the SCIATRAN radiative transfer model are used as described by Noël et al. (2024). Here, we provide a brief summary of the dataset.

For this dataset, eight ACT (approximately every fifteenth) and every twentieth ALT spatial samples with solar zenith angles (SZAs) smaller than $80°$ (consistent with ESA, 2020) were selected using CO2M orbit data of one year provided by EUMETSAT to simulate nadir radiances over land at these CO2M spatial samples with SCIATRAN (Rozanov et al., 2017). The subset is chosen to reduce computation time while keeping an annual dataset and representatively sampling the geophysical conditions. The SCIATRAN radiative transfer model can be used to simulate radiative transfer through the Earth's atmosphere, including multiple scattering, in a wide range of wavelengths. For the generation of the dataset for this study, input pressure, temperature, clouds and water vapor profiles have been taken from the ECMWF re-analysis version 5 (ERA5, Hersbach et al., 2020) and other trace gas profiles as well as input parameters for the simulation of aerosols are taken from the Copernicus Atmosphere Monitoring Service (CAMS, Inness et al., 2019)) re-analysis data, both from the reference year 2015. The surface reflectivity needed to calculate the radiances have been derived using satellite measurements of the Moderate Resolution Imaging Spectroradiometer (MODIS). The simulation of solar chlorophyll fluorescence is based on Rascher et al. (2009). The simulations are restricted to scenes over land in nadir geometry. Further details can be found in Noël et al. (2024).

## 4 Defining scenarios avoiding detector saturation

The goal of this section is to determine the maximum radiance in each spectral band and compare it with the OSF-specific saturation limit in order to estimate the OSF needed to avoid saturation. While different OSFs could be used along one orbit,



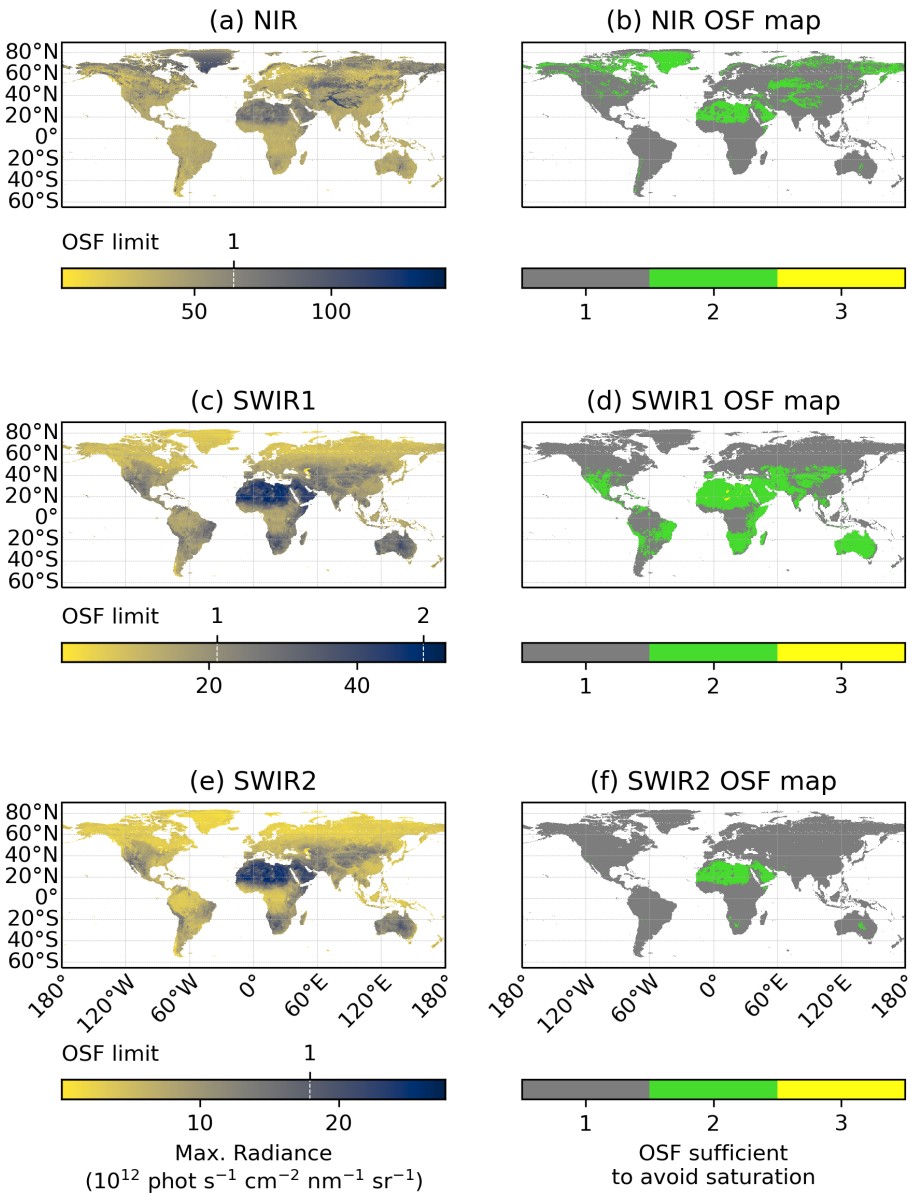

**Figure 2.** Left column: spectral and temporal maximum radiance in the simulated 1-year subset nadir radiances, binned to a 0.4x0.4 degree latitude-longitude grid. Clouds are removed for this analysis. The radiance limits corresponding to the OSFs in each wavelength band are illustrated by white dashed lines in the color scales of the panels. Right column: OSF needed in order to avoid saturation for the maximum radiances. The rows show the wavelength bands of CO2I: a, b NIR; c, d SWIR1 and e,f SWIR2. The simulated radiances are for nadir over land, only. Therefore, the ocean is masked by white colors.




**Table 1.** Fraction of cloud-free spatial samples (in %) for which the OSF in the first column is needed to avoid saturation in the 1-year dataset of simulated radiances.

| OSF | NIR | SWIR1 | SWIR2 |
|-----|-------|-------|-------|
| 1 | 93.01 | 75.81 | 94.36 |
| 2 | 6.99 | 24.15 | 5.64 |
| 3 | 0.00 | 0.04 | 0.00 |

switching would make mission operations more complex and may lead to other challenges such as OSF-dependent calibration (cf., Kataoka et al., 2017). Here, we investigate whether it is possible to use constant OSF settings all over the globe.

The left column of Fig. 2 shows the spectral and temporal maximum radiance occurring during the 1-year dataset of sim-
ulated radiances. They are binned to a 0.4x0.4 degree grid which corresponds roughly to the distance of every 15th spatial sample of CO2I on the Earth's surface. As expected due to the solar irradiance spectrum, radiances are larger in the NIR than in the SWIR bands. Largest maximum radiances are simulated over the desert regions like the Sahara and the Australian deserts, especially in the SWIR bands which are sensitive to different surface types (e.g., Fasnacht et al., 2019; Manakkakudy et al., 2023; Santamaría-López et al., 2024). Increases also occur over the tropical rain forests due to the red-edge of plants
(e.g., Ge et al., 2019; Zeng et al., 2021). The ice-covered surface of Greenland shows increases in NIR and small values in the SWIR bands.

The color scales in Fig. 2 also include the radiance limits for the OSFs in each band. The global maximum radiance in NIR and SWIR1 correspond to values exceeding the limit of saturation for OSF 1, indicating that OSF > 1 might be needed to avoid saturation. The right column of Fig. 2 shows which OSF is sufficient in each grid box. The limit for OSF 1 is exceeded
over most parts of the land surface in the NIR band. In SWIR1, the latitude regions roughly between 30° N/S that include the tropical forests and deserts are also exceeding the saturation limit for OSF 1. The only region where an OSF of 3 is needed in the SWIR1 band is in the middle of Sahara where no significant anthropogenic sources of greenhouse gases exist. Therefore, OSF 1 and 2 are the dominant OSFs that are sufficient for SWIR1. In SWIR2, only some of the deserts show exceedance of the limit for OSF 1 where then OSF = 2 would be necessary to avoid saturation.

Table 1 shows the global fraction of spatial samples in the one-year subset dataset where the shown OSF is needed to avoid saturation. As can be seen, about 7 % of all spatial samples within the simulated year exceed the threshold for OSF 1 in the NIR band. Although some spatial samples showed exceedance of the threshold for OSF = 2 in the NIR band (Fig. 2), the actual fraction in the whole one-year dataset is negligible. In the SWIR1 band, about 24 % of all spatial samples require OSF = 2. The locations requiring an OSF of three (yellow in panel d of Fig. 2) correspond only to a minor fraction of 0.04 %. As expected
from the previous analysis, the fraction for OSF of two is smaller for SWIR2 than for SWIR1 with a value smaller than 6 %.

Therefore, a significant fraction of spatial samples exist that are affected by saturation with OSF = 1 in all bands so that OSFs larger than one should be further investigated which is the subject of this study. As the main fraction of saturated spatial samples are located over regions that are not known for large emission $CO_2$ sources, like over deserts and snow, we consider





**Table 2.** Oversampling factor (OSF) scenarios for the spectral bands of CO2M with their notation in this study. These OSFs are assumed to be applied globally in this study, details see text.

| Notation | $OSF_{NIR}$ | $OSF_{SWIR1}$ | $OSF_{SWIR2}$ |
|---|---|---|---|
| 111 | 1 | 1 | 1 |
| 222 | 2 | 2 | 2 |
| 232 | 2 | 3 | 2 |
| 333 | 3 | 3 | 3 |

the OSF scenario OSF = 1 in all wavelength channels not only as a reference but also as one of the likely scenarios for CO2M.

In addition, scenarios with OSF = 2 in all channels is considered in this study. As we use only a sub-sampled dataset for the global analysis we also add an OSF scenario with OSFs of 2 (NIR), 3 (SWIR1) and 2 (SWIR2) and a scenario with an OSF of 3 in all bands in case the missing spatial samples show larger radiances. The scenarios are summarized in Table 2 and denoted as OSF 111, 222, 232 and 333, respectively.

## 5   The FOCAL greenhouse gas retrieval algorithm

In this study, we use the updated version 1.1 of FOCAL-CO2M which is similar to the version used by Noël et al. (2024) with minor updates of coding optimizations. Therefore, we provide a brief summary of FOCAL here, with further details to be found in Noël et al. (2024) and references therein.

FOCAL is a radiative transfer and trace gas retrieval code approximating scattering in the atmosphere by a single scattering layer whose height, optical thickness and Ångström exponent are retrieved as part of the algorithm using optimal estimation.

This approximation leads to an analytic expression for the calculation of scattering (Reuter et al., 2017b) making FOCAL a fast algorithm for the inversion of greenhouse gas concentrations from spectral measurements in the NIR and SWIR. FOCAL has been successfully applied to many satellites measuring greenhouse gases, such as OCO-2 (Reuter et al., 2017a, b) and GOSAT and GOSAT-2 (Noël et al., 2021, 2022), and is one of the three operational algorithms to retrieve greenhouse gases from the future CO2M mission.

The FOCAL algorithm comprises pre-processing (i.e. filtering of measurements with bad quality and difficult scenes such as high SZAs), inversion and forward model (i.e. optimal estimation with an iterative approach starting with a-priori knowledge) and post-processing (i.e. convergence and variance filtering and bias correction). The setup of these steps here is similar to that used by Noël et al. (2024), which is why we refer to this publication for the details and describe here the adaptions made for this study.

As the noise model and the post-processing are specific to the setup of the instrument, e.g. the OSF, and we assume the application of one OSF scenario all over the globe, we use different noise models and post-processing for each OSF scenario. The post-processing uses a variance minimization process and filters data that have the largest impact on the scatter of the





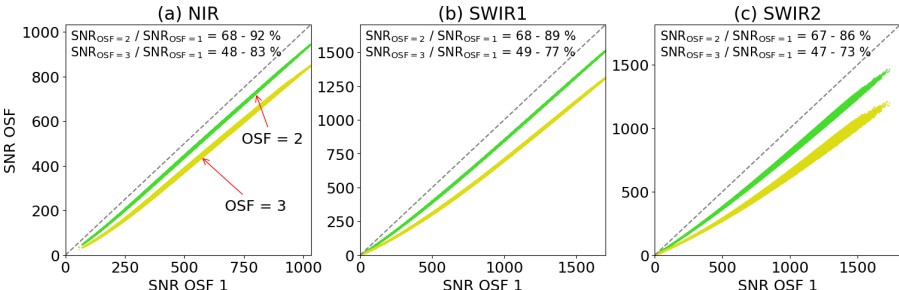

**Figure 3.** Signal-to-noise ratios compared to an OSF of one in (a) the NIR, (b) the SWIR1 and (c) the SWIR2 bands. The black dashed line illustrates the one-to-one line. In addition, the minimum and maximum of the ratio between the SNR to SNR for $OSF = 1$ is labeled in each panel for each OSF.

difference to the truth so that, in average, the differences to the truth are minimized. This means that variables and their limits to minimize the variance differ between the OSF scenarios, while keeping the total fraction of measurements that are removed,

which makes the OSF scenarios comparable to each other. Note that post-processing is based on $10\,\%$ of the whole year's data. The setup of the noise model and the post-processing can be found in Appendix A.

In this study, we apply FOCAL-CO2M version 1.1 to the one year of simulated subset nadir radiances over land, filtered for clouds, a SZA larger than $75°$ and saturation, and retrieve $XCO_2$ using the OSF scenarios of Table 2. The impact on $XCO_2$ is discussed in the following section.

We will use the retrieval's random noise error arising from the different components discussed in Sect. 2 to investigate the impact on the $XCO_2$ precision. In addition, we will analyze the retrieval's smoothing error, which arises from the smoothing of the state connected to the averaging kernels (Rodgers, 2000), in combination with the noise error to make statements about the impact on the overall noise error. Remaining systematic errors after post-processing due to the different OSF settings should be small because the post-processing is calculated individually for each OSF scenario and will be analyzed in terms of overall

standard deviations of the retrieval residuals.

## 6 Impact of increasing the OSF

### 6.1 Impact on SNR

We first analyze how the SNR of the continuum radiance is affected by an increased OSF. Figure 3 shows the reduction of the SNR compared to $OSF = 1$ for the three spectral bands. While the maximum SNR for $OSF = 1$ is about 1000 in the NIR and

1600 in the SWIR bands, it is smaller for larger OSFs. For large SNRs, i.e. large radiances, $A$ dominates Eq. (4) leading to a constant slope in all cases. The non-linear part corresponding to $B$ leads to changes of the slope at the lower end of SNRs so that all lines converge to the origin. Ratios of SNR to OSF 1 are printed for each spectral band and are between 67 and $89\,\%$ for $OSF = 2$ and between 47 and $83\,\%$ for $OSF = 3$ in all bands, see Fig. 3.





In the next step, we investigate the impact of the SNR changes, made in the previous analysis, on the retrieved a posteriori noise of FOCAL. This could be used in the future to estimate the $XCO_2$ precision with the knowledge of the SNR change. In the easiest case, relative changes of the SNR ($\Phi$) are proportional to relative changes of the a posteriori noise error ($N$), which can be written as

$$\frac{dN}{N} = C\frac{d\Phi}{\Phi} \tag{7}$$

with $C$ as constant translating SNR changes to changes in $XCO_2$ a posteriori noise, i.e. precision. This assumption is tested in this section. Integrating Eq. 7 yields

$$\ln N = C \cdot \ln(\Phi) + k, \tag{8}$$

where $k$ is an integration constant and the knowledge that $N$ and $\Phi$ are positive numbers has been applied.

Equation 8 describes a linear relationship between N and $\Phi$ in a double-logarithmic space. It has been tested for all scenarios and bands which are shown in Fig. 4. This figure shows histograms of binned logarithmic SNR and a posteriori noise error values with linear regressions as dashed lines and formulas in the respective legends. The assumption of linearity does not apply to the NIR band because some high SNR values also have a large noise error. On the other hand, there is a clear linear relationship in the two SWIR bands. While the slopes of the regression lines differ among the OSF scenarios the negative values are largest in the NIR (average -0.69), smaller in SWIR1 (average -0.62) and smallest in SWIR2 (average -0.45). The smaller slope in SWIR2 compared to SWIR1 can probably be explained by the different number of spectral detector pixels in the $CO_2$ absorption region within the FOCAL fit windows: about 473 in SWIR1 and 770 in SWIR2 which makes single noisy measurements less sensitive to changes in $XCO_2$ in SWIR2. In addition, the sensitivity of the absorption lines to $CO_2$ changes are different in SWIR1 and SWIR2.

In summary, the SWIR2 band is less sensitive to changes in the SNR than SWIR1 where both the double-logarithmic linear relationship could be confirmed. Apart from values with high SNR and large noise for which the assumption of linearity does not hold, the slope in the NIR band is similar to that in the SWIR1 band, suggesting that the NIR band is of similar importance for the retrieval of $XCO_2$.

## 6.2 Impact on coverage

Tests showed that if saturation occurs it usually does not happen only at one spectral detector pixel but for more than 60 pixels. Therefore, it can be expected that a large fraction of the continuum range of the spectrum is affected by saturation so that the measurement is not useful for the retrieval and has to be deleted from the record which will reduce the coverage on Earth. In order to analyze the impact of filtering for saturation, the simulated radiances are binned to a 0.4x0.4-degree latitude-longitude grid and the fraction of remaining data is shown in the first column of Fig. 5. As discussed above, the desert regions have large reflectances that will lead to saturation for all measurements in OSF 111 (panel a) e.g. over the Saharan region, the Arabian Peninsula and the deserts in Australia. In addition, the surface covered by ice such as Greenland and the Himalayas are filtered out as a result of saturation using OSF 111. In total, a fraction of 72.7 % remains globally when adding an additional pre-processing filter for saturation in OSF 111.



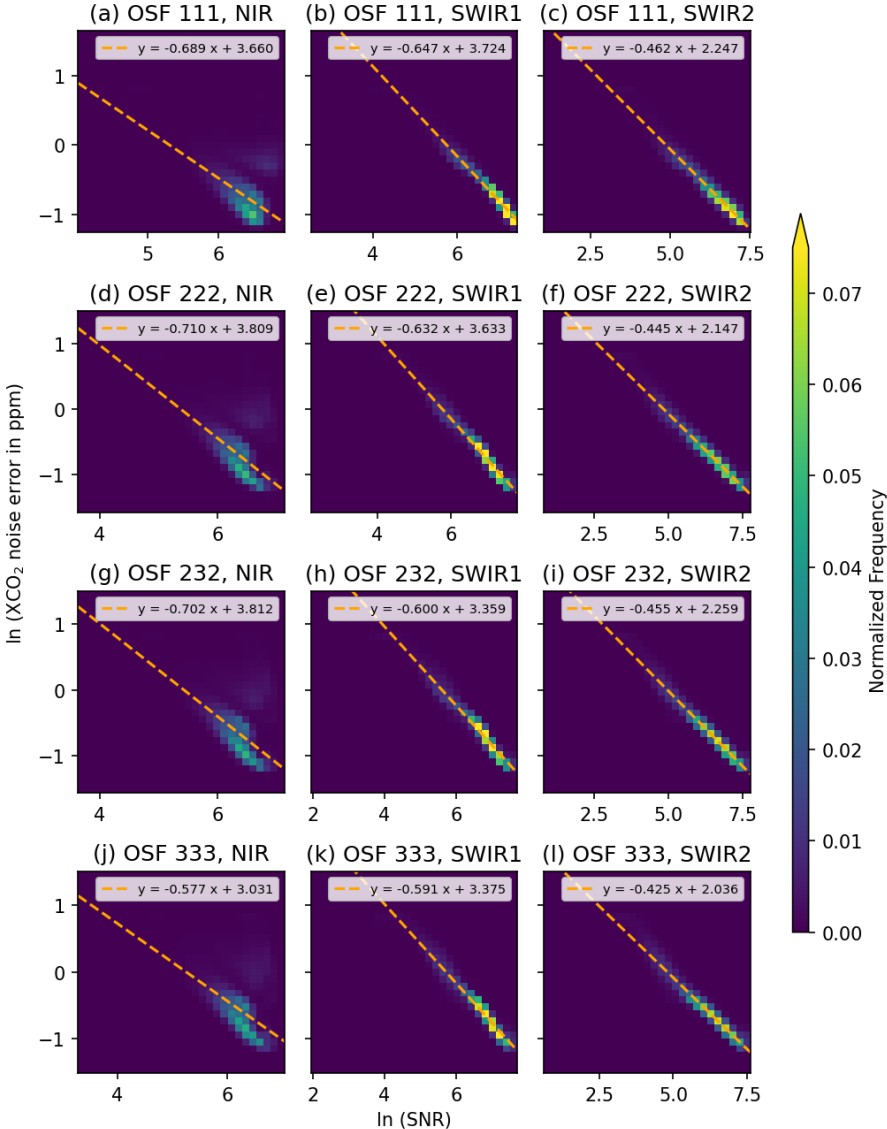

**Figure 4.** Logarithm of the continuum SNR versus logarithm of the FOCAL $XCO_2$ a posteriori noise error for all combinations of OSF scenarios and wavelength bands. Orange dashed lines show linear regressions for each panel with its parameters in their legend. Note the different values on the x-axes of each panel.

As expected from the analysis of Sect. 4, increasing the OSF to values larger than one leads to better coverage. For instance, with OSF 222 (panel c), saturation only occurs at localized spots on the Sahara leading to an overall remaining fraction of data larger than 99 %. For the scenarios with larger OSFs (232 and 333), 100 % of the values remain, i.e. no saturation or saturation in the sub-% range is simulated in these cases. On the other hand, the majority of locations affected by saturation are in regions





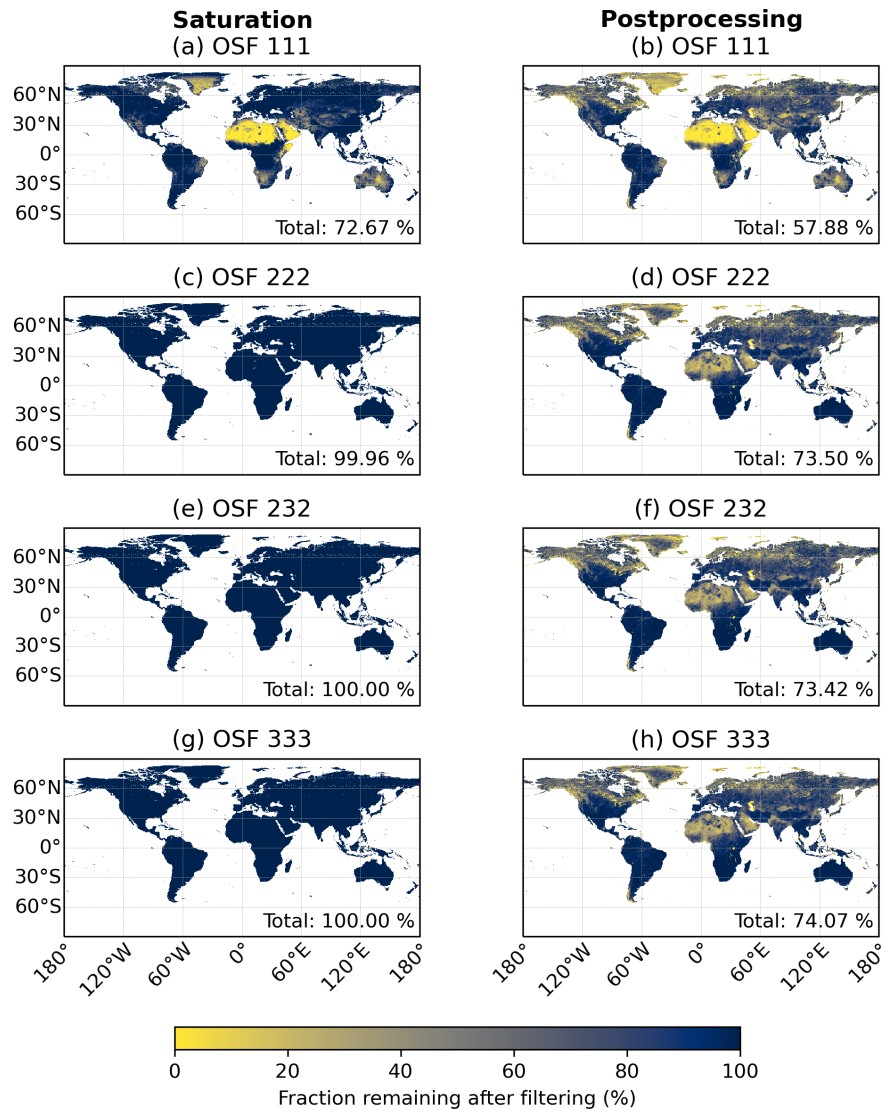

**Figure 5.** Throughput of the saturation filter during pre-processing (left column) and after post-processing (right column).The data are binned to a 0.4x0.4-degree latitude-longitude grid based on the CO2M spatial sample center coordinates. Note that the saturation filter is applied as the last filter and $100\,\%$ means the data after filtering for SZA $> 75°$ and cloud fraction $> 0.2$. The global fraction of remaining data is labeled as "Total" in the panels. The filters applied during post-processing can be found in Table A1 of Appendix A.

where no significant emission sources exist so that OSF 111 could still be sufficient for the goal of estimation of localized emission sources.

Another aspect is post-processing which filters parts of the data, independent of saturation, and the fraction of data remaining after saturation filter and post-processing is shown in the second column of Fig. 5. The filters applied during post-processing



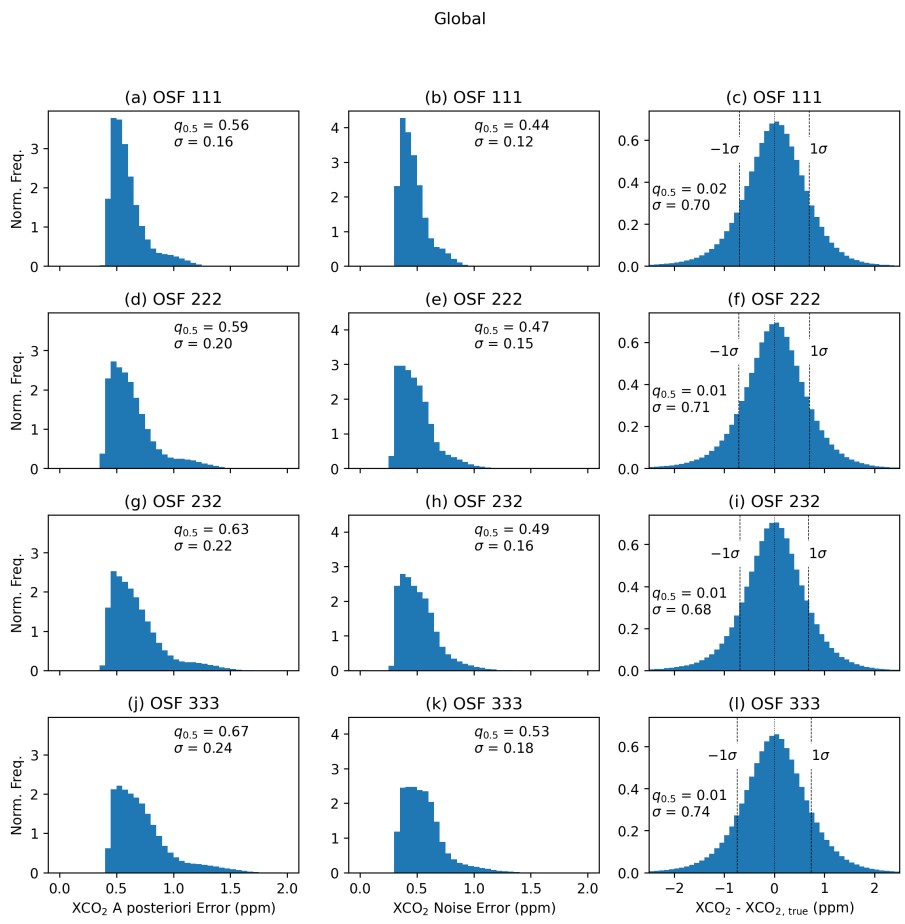

**Figure 6.** Histograms of the global distribution of the a posteriori $XCO_2$ retrieval error (first column), the a posteriori $XCO_2$ noise error (second column) and the differences between FOCAL $XCO_2$ after post-processing and the true $XCO_2$ (third column) for the whole year of simulated subset radiances. The rows show OSF 111, 222, 232 and 333, respectively. Median ($q_{0.5}$) and standard deviation ($\sigma$) of the distributions are added to each panel in units of ppm. Note that the standard deviation of $XCO_2 - XCO_{2,\text{true}}$ includes both systematic and random noise contributions.

depend on the scenario and can be found in Table A1 of Appendix A. While the post-processing filters part of the data over the Sahara and the Arabian Peninsula also for $\mathrm{OSF} > 1$, data are additionally lost at the high latitudes. The patterns are similar for all OSF scenarios. In total, about $58\,\%$ are left for OSF 111 and about $73\,\%$ for the other scenarios.

## 6.3   Impact on $XCO_2$

While the overall coverage increases with OSFs larger than one the precision is reduced due to read-out noise. We determine
the impact on the precision by calculating FOCAL's a posteriori $XCO_2$ retrieval noise. Histograms of the a posteriori error



(which is the root sum squares of the noise and smoothing errors), noise error and the difference of retrieved $XCO_2$ minus a priori (equal to true) $XCO_2$ for the whole one-year dataset and all OSF scenarios of Table 2 are shown in Fig. 6. All panels include median, $q_{0.5}$, and standard deviation, $\sigma$, of the respective histogram. Both a posteriori error and noise error increase with larger OSFs. For OSF 333 errors are estimated to be a factor of about 1.2 larger than the OSF 111 errors in the median.

For OSF 222, this factor is 1.06. Note that this value refers to the median noise and can be larger for single measurements, see distributions of middle column in Fig. 6. A discussion where the impact on precision is largest for the respective OSF scenarios follows later in this section. In addition, the variance, which is the square of these values, is connected to the information reduction, which is 39 % for OSF 333 and 12 % for OSF 222, compared to OSF 111.

The median of the a posteriori error is $0.56$ ppm for OSF 111 compared to a $0.44$ ppm noise error and similar for the other
OSF scenarios. Therefore, the error of $XCO_2$ is dominated by the noise component of the error induced by changing the OSF in each scenario.

As expected from adapting the post-processing to each OSF scenario, the distributions of $XCO_2 - XCO_{2,\text{true}}$ in the right column of Fig. 6 are nearly symmetric around $0$ ppm, as demonstrated by median values close to zero. Note that the post-processing is based on $10$ % of all available data. Due to slightly larger noise errors for $OSF > 111$, the standard deviations
increase slightly from $0.70$ ppm for OSF 111 to $0.74$ ppm for OSF 333. Note that this value includes both systematic and noise errors that was not done in this analysis.

The OSF scenarios 111 and 222 show an increase of median noise errors in the order of $0.03$ ppm so that the decrease in global $XCO_2$ precision is estimated to be small in the setup of this study.

We also analyzed the monthly evolution of the error and the number of spectral samples for all OSF scenarios after post-
processing, see Fig. 7. Timeseries of the global monthly median $XCO_2$ noise error for all OSF scenarios can be found in panel a. The noise increases by a constant factor between scenarios OSF 222, 232 and 333. This is different for OSF 111 because the number of data (panel b) is decreased by about $20$ % in this scenario. The noise error shows a semi-annual cycle with maximum values in June and December, the respective summer months on the hemispheres. These peaks can be seen in the individual latitude bands shown in the rows of Fig. 7. In addition, most of the data loss in the OSF 111 scenario occurs in
latitudes between $40°$S and $40°$N where most of the deserts are located, consistent with the previous findings. In all other latitudes, the median $XCO_2$ noise error increases with increased OSF in the SWIR bands, which are sensitive to changes in $CO_2$. The results of the northern mid-latitudes in Fig. 7e are the best approximation of the VEG50 scenario which is based on typical mid-latitude vegetation conditions like albedos and a solar zenith angle of $50°$ and which defines the requirements for CO2M (ESA, 2020). Tests with VEG50 (not shown) showed similar values of the $XCO_2$ noise as in the winter months of the
panel: approx. $0.7$ ppm for OSF 111, $0.77$ ppm for OSF 222 and about $0.88$ ppm for OSF 333. Therefore, these results are consistent with this experiment.

As calculations of emissions depend linearly on the $XCO_2$ enhancement to some background value (e.g., Fuentes Andrade et al., 2024), the error of emission estimates scales linearly with the $XCO_2$ a posteriori noise of single soundings. Thus, the relative change in $XCO_2$ a posteriori noise is the same for uncertainties in the emissions. This was tested with a simple emission
model in the scope of this study. As an example, using mid-latitude summer conditions where the median $XCO_2$ a posteriori



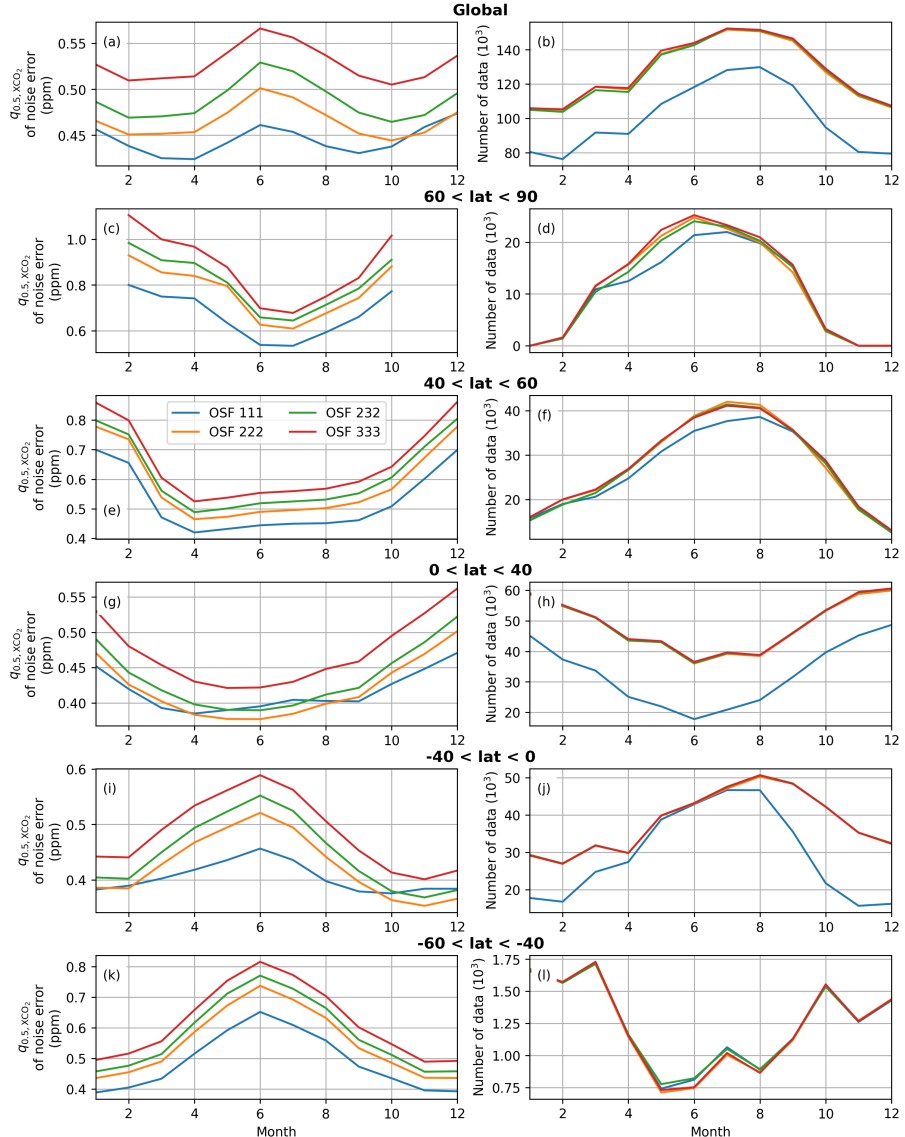

**Figure 7.** Timeseries of the monthly values for various latitude bands (rows) of the following for all OSF scenarios of Table 2: The panels in the left column show the monthly median ($q_{0.5}$) of the $XCO_2$ noise error. The panels in the right column illustrate the number of data that are left after filtering for saturation in the pre-processing and after post-processing, i.e. data that have converged and are not filtered out during post-processing. In the latitude band between 60 and $90°$S, data coverage is small so that it is not shown here.

noise error increases from $0.45\,\mathrm{ppm}$ (OSF 111) to $0.5\,\mathrm{ppm}$ (OSF 222), see panel e of Fig. 7 in June, it can be expected that the relative increase in the uncertainty of the emission due to noise is the same: 1.11 in the median which can be larger for individual emission estimates.



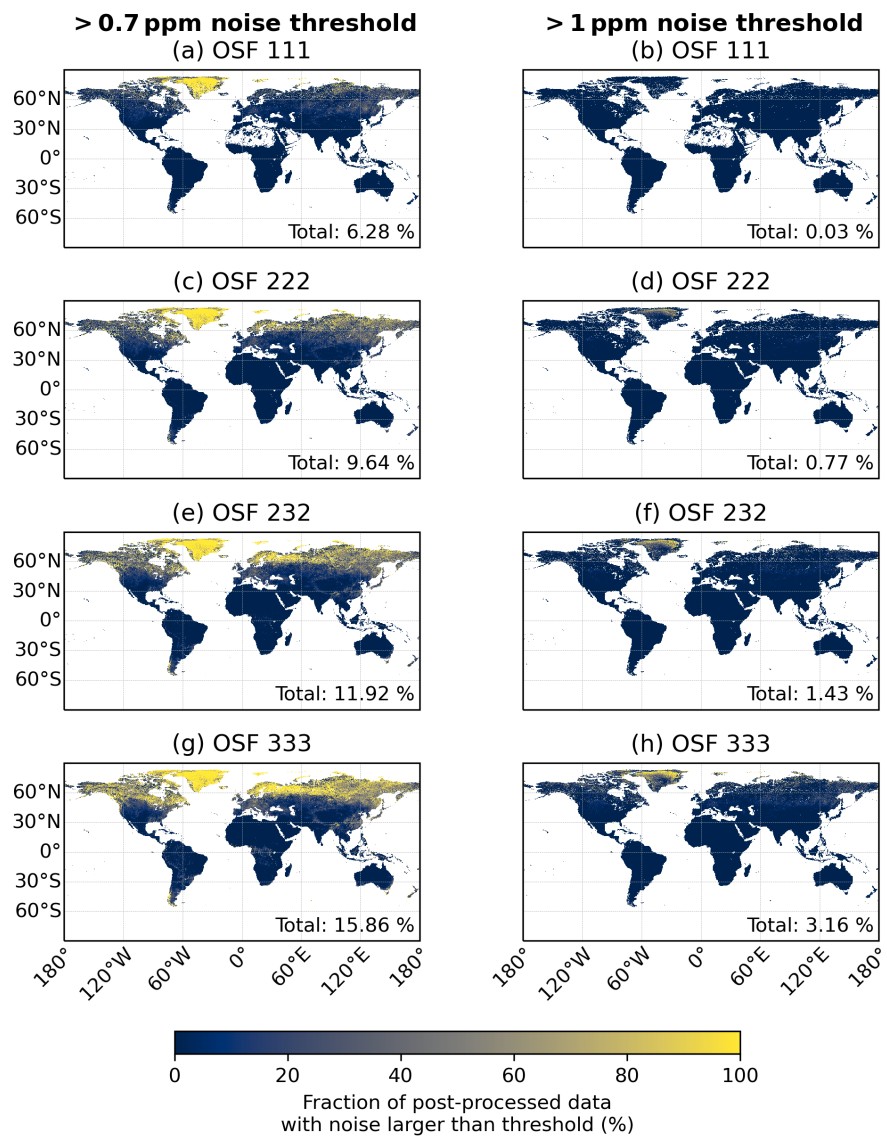

**Figure 8.** Fraction of spatial samples whose $XCO_2$ a posteriori noise error is larger than $0.7\,\mathrm{ppm}$ (left column), based on the precision requirement in ESA (2020), and $1\,\mathrm{ppm}$ (right column). The data are binned to a 0.4x0.4-degree latitude-longitude grid based on the CO2M spatial sample center coordinates. Note that the fraction is related the data after postprocessing in this figure. The global fraction is labeled as "Total" in the panels.

For the estimation of emissions, high precision of the measurements is needed which decreases with larger OSF. Therefore,
the columns of Fig. 8 show the fraction of post-processed data that are affected by a precision threshold of $0.7\,\mathrm{ppm}$ and $1\,\mathrm{ppm}$. This mostly affects the Northern mid and high latitudes. Most of the post-processed data have an a posteriori noise error





smaller than $1\,\mathrm{ppm}$. The global fraction of affected spatial samples is smaller than $3.2\,\%$ for all OSF scenarios and is largest on Greenland where no significant sources of $CO_2$ exist. Furthermore, the possibility of glint mode could change these results especially over the snow-covered regions on Earth because snow has generally an enhanced forward scattering component

(e.g., Mikkonen et al., 2024). On the other hand, $6\,\%$ of the post-processed data have a noise error larger than $0.7\,\mathrm{ppm}$ in OSF 111, about $9\,\%$ in the OSF scenario 222, more than $11\,\%$ in OSF 232 and about $16\,\%$ in OSF 333. Hence, estimates of emissions over Scandinavia, Canada and Northern Russia might be difficult when using an OSF larger than 2 because of the precision degradation there.

In summary, these results imply that increasing the OSF globally to values larger than one leads to minor decreases in the

global median precision while increasing the coverage. Based on the simulated one-year subset of radiance data used here, the scenarios 111 and 222 seem to be favorable for CO2M in the future: Although OSF 111 will lead to saturation over the deserts and some snow-covered regions, the precision impact is smallest among the scenarios which might be of importance for the mid-latitudes. With the OSF scenario 222, we found nearly global coverage with respect to saturation, but a larger impact on the precision in the mid and high latitudes.

## 7  Summary and Conclusions

The Copernicus Anthropogenic $CO_2$ Monitoring (CO2M) mission is a satellite constellation with the first satellite expected to be launched in 2026. One of the operational greenhouse gas retrieval algorithms for CO2M will be the Fast atmOspheric traCe gAs retrievaL (FOCAL) algorithm. In this study, we analyzed the impact of scenarios avoiding saturation of the detector which may occur for instance at bright scenes on the Earth's surface and which have to be filtered out during the retrieval

so that the coverage is decreased. This can be avoided by increasing the number of read-outs per integration time step, i.e. increasing the oversampling factor (OSF) which, on the other hand, decreases the SNR of the measurement. We used idealized simulated radiances for this study with the goal being to investigate the long-term impact of using different OSFs. This was done by examining spatial-sample-wise saturation without considering any effect from either nearby bright scenes from the surrounding spatial samples or from nearby bright scenes outside the swath that could lead to saturation as well, such as stray

light from nearby clouds. Clouds can lead to saturation as well so that the OSF might have to be increased for that scene.

We used a one-year subset dataset of simulated radiances for conditions of 2015 to define scenarios of OSFs and then to investigate the impact on $XCO_2$ using FOCAL. The post-processing was adapted depending on the used OSF because changing the OSF changes the retrieval-related detector properties. Our assumption was to keep one OSF setting for the whole globe and year to avoid possible calibration difficulties due to changing OSFs during the operation and to keep the operation of the

satellite as simple as possible.

We compared the maximum radiances with the OSF-specific maximum radiance that can be detected in all spectral bands in order to define scenarios of OSFs to be used in the long-term analysis. We found that saturation especially occurs over deserts and the parts covered by rocks. Scenarios increasing the OSF for all CO2M spatial samples to values between two and three in



the near-infrared (NIR) and short-wave infrared (SWIR) bands were defined which were called OSF 111 (baseline), 222, 232
and 333 in the order of OSFs set in the NIR, SWIR1 and SWIR2 bands, respectively.

We found that the decrease in signal-to-noise ratio (SNR) due to different OSFs has a wavelength-band-dependent impact on the $XCO_2$ a posteriori noise error, with smallest sensitivity in the SWIR2 band. The impact on $XCO_2$ was analyzed with distributions of the global noise error estimates which increased by a factor of 1.18 between OSF scenarios 111 and 333 and 1.06 between OSF 111 and 222, which in the median is not large but which leads to precision degradation in the Northern
high-latitudes at the edge of emission estimation. The results showed that the filtered regions mostly include regions that are not known to have large emission sources. Therefore, the degradation of precision in these regions might be acceptable so that the assumption of a uniform OSF might also be relaxed and the OSF could be switched to OSF 222 over regions like the deserts and OSF 111 elsewhere.

These results are based on idealized simulation of surface properties and assuming a perfect instrument. Thus, errors might
be larger for real measurements and the results shown here can only provide a first insight towards the actual impact of changing the OSF when applied to real measurements. The analysis is limited to the nadir configuration over land and further investigations for other forseen geometries like ocean glint are needed in the future. In addition, the impact on emissions, especially in the northern high latitudes such as Scandinavia, Russia and Canada, should be further investigated in more detail in the future. While the analysis of the saturation filtering is independent of the retrieval method used, the further results will
depend on the retrieval algorithm and are likely to be different for Fusional-P-UOL-FP and RemoTAP. As discussed e.g. by Noël et al. (2024), the requirements concerning $CH_4$ are not as stringent as for $CO_2$ which is why we restricted the analysis to $CO_2$ in this study.

Overall, we found increases in the coverage when using OSFs larger than one and decreases in precision. Based on the idealized simulated radiances, the scenarios OSF 111 and 222 could be considered as possible OSF scenarios for CO2M for
$XCO_2$ and under the assumption of a fixed OSF setting independent of the location.

*Data availability.* The data used in this study is available by the authors on request.

## Appendix A: Retrieval setup for the OSF scenarios

As discussed in Sect. 5, the retrieval setup depends on the OSF scenario. For the forward model error, we strictly followed the concepts outlined by Reuter et al. (2017a) and Noël et al. (2024). Table A1 summarizes the variables used and the corresponding
limits of the variance filter applied during post-processing.



**Table A1.** OSF-dependent limits of retrieval variables applied during post-processing to minimize the overall variance, sorted by their relevance. The variables are albedo coefficients $A_{i,band}$ of second-order polynomial and the cost function $\chi^2$. No limit is denoted as dash. Note that the variable notation comes from Reuter et al. (2017a) and is not to be confused with the parameter $A$ of the SNR in the main text.

| OSF Scenario | Parameter | lower limit | upper limit |
|---|---|---|---|
| 111 | $A_{3,\text{SWIR1}}$ | $-2.5619 \cdot 10^{-5}$ | – |
| | $A_{0,\text{SWIR1}}$ | $0.1115$ | – |
| | $A_{3,\text{SWIR2}}$ | – | $3.6136 \cdot 10^{-5}$ |
| | $\chi^2$ | – | $1.0669$ |
| 222 | $\chi^2$ | – | $1.0568$ |
| | $A_{0,\text{NIR}}$ | $0.1037$ | – |
| | $A_{0,\text{SWIR1}}$ | $0.1100$ | – |
| | $A_{3,\text{SWIR2}}$ | – | $5.1109 \cdot 10^{-5}$ |
| | $A_{2,\text{SWIR1}}$ | $-4.1952 \cdot 10^{-5}$ | – |
| | $A_{2,\text{SWIR2}}$ | – | $3.5254 \cdot 10^{-5}$ |
| 232 | $A_{2,\text{SWIR1}}$ | $-4.3646 \cdot 10^{-5}$ | – |
| | $A_{2,\text{SWIR2}}$ | $-0.0002$ | $3.1977 \cdot 10^{-5}$ |
| | $A_{0,\text{SWIR1}}$ | $0.1127$ | – |
| | $\chi^2$ | – | $1.0282$ |
| 333 | $A_{2,\text{SWIR1}}$ | $-4.5681 \cdot 10^{-5}$ | – |
| | $A_{0,\text{SWIR1}}$ | $0.1090$ | – |
| | $A_{3,\text{SWIR2}}$ | – | $6.7002 \cdot 10^{-5}$ |
| | $\chi^2$ | – | $1.0026$ |

*Author contributions.* MW carried out the analysis and wrote the initial draft of the manuscript. YM provided the data about the oversampling setup. SN, MR and MH generated the one-year subset dataset of simulated radiances for this study. All authors contributed to write the manuscript.

*Competing interests.* The authors declare that they have no competing interests.

*Acknowledgements.* We acknowledge funding by the ESA CO2M Science Study under contract no. 4000138164/22/NL/SD, lead by SRON Netherlands Institute for Space Research. Parts of this work have been carried out with funding by the European Union Copernicus programme through EUMETSAT contract EUM/CO/19/4600002372/RL. Parts of this work are funded by the German Federal Ministry of Education and Research (BMBF) project "Integrated Greenhouse Gas Monitoring System for Germany – Observations (ITMS B)" under



grant number 01 LK2103A and the State and the University of Bremen. All calculations reported here were performed on HPC facilities of
the IUP, University of Bremen, funded under DFG/FUGG grant INST 144/379-1 and INST 144/493-1.



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
