# Peer review of "A study of measurement scenarios for the future CO2M mission: avoidance of detector saturation and the impact on XCO2 retrievals"

_EGUsphere, 2024_

## Referee Comment (RC1)

**Review of "A study of measurement scenarios for the future CO2M mission: avoidance of detector saturation and the impact on XCO2 retrievals"**

Comments based on https://egusphere.copernicus.org/preprints/2025/egusphere-2024-3857/
Preprint, retrieved 10 February 2025

**General comments**

Dear authors,

The paper is in general written in a clear and succinct manner and the reasoning is good to follow. The subject is highly relevant for the future CO2 mission.

There are however a few points, where clarifications are needed and the consistency needs to be checked. The biggest question I have is (see also SC2), if – instead of using oversampling higher than one – the sampling time can be decreased? Has this been investigated? Now the investigation is reduced to integration times of 271, 117 and 66ms. What would for example 250 ms sampling time produce as a result? Then the ground sample would be about 1.8 x 1.8 km, so the ground resolution would be higher (and produce symmetric ground pixels). What are the reasons to use 271 ms? With a back of the envelope calculation (using the results from Table 1) I estimate that 208 ms integration time are needed that the 24% saturated pixels for the OSF=1 case are within the FWC limit. Are there limitations on datavolume, internal datarate, synchronization of read-outs, co-registration or other reasons to use 308ms sampling time? Is there an estimate of the level of saturation versus integration time? How large are the signal independent contributions (dark current, thermal background, offset)? Can they be reduced? Please address these points in the article.

Please find below further specific comments on the content and in a separate table technical comments on typos and phrasing.

**Specific comments**

| Item | Section | Line | Comment |
|------|---------|------|---------|
| SC1 | 2 | 86 | CO2I -> CO2I/NO2I: the slit is also shared with the VIS spectrometer, so CO2I alone is not accurate |
| SC2 | 2 | 92 | As an alternative to the OSF couldn't the sampling time of 308ms be reduced? With 250 ms the ground sample would be about 1.8 km x 1.8km. Why not? What are the limitations (detector limitations, datavolume,...)? Is there an estimation how short the exposure time would have to be to avoid saturation everywhere? See also separate discussion above. |
| SC3 | 2 | 110 | The spatial sampling is not affected, but I would expect an impact on the the spatial energy distribution function. Can you please comment on this? |
| SC4 | 2 | 114 | Only the detectors for SWIR 1 and 2 are mentioned here. What about NIR? What is the FWC of the NIR detector? Is it the same? |

| | | | |
|---|---|---|---|
| SC5 | 2 | 118 | "a radiance spectrum with saturated pixels has to be discarded" The reasoning to discard the entire spectrum is not described clearly. I would advice to change the order of this paragraph somewhat and mention firstly (as described in line 268) that not single pixels but at least 60 are affected (what fraction is that of the spectrum?) and secondly that this impacts the straylight correction. |
| SC6 | 2 | 126 | "neglect the effect... on neighbored spatial samples" . This is unclear: do you mean other viewing angles/swath angles/ spatial samples in the same frame where saturation occurs? Then it should be excluded as a whole anyways, as the straylight correction would be insufficient. Or do you mean the impact on following read-outs? (see next comment). |
| SC7 | 2 | 127 | Is there anything known about detector blooming or the effect of pixelsaturation on the following (unsaturated) read-outs? Is there a memory effect? Or is the assumption here that only an individual frame is affected? |
| SC8 | 3 | 168 | Can something be said about the off-nadir angles? Is the effect of saturation expected to be smaller there? And do you then assume the nadir spectrum for all viewing angles (spatial samples on the detector). This sentence seems to contradict the statement in line 126. |
| SC9 | 6.1 | 242 | The numbers in the text are not consistent with the insets in Fig. 3. If you want to give ranges which include all bands, it should be 67 and 86% and 47 to 73 % for OSF 2. Or you can remove "in all bands" in line 243 and use the ranges 67-92% and 47-83% |
| SC10 | 6.1 | 256 | "some high SNR values have a large noise error" Could you please add an explanation why that is the case? |
| SC11 | 6.3 | 306 | "not done in this analysis": was this shown elsewhere? Please add a reference. |
| SC12 | 6.3 | 312 | "decreased by about 20%", is this due to the saturation filtering? Please clarify this in the manuscript. |
| SC13 | 6.3 | 333 | "glint mode could change" , change in what way? Please specify what you expect. |

**Technical comments/typos**

| Item | Section | Line | Comment |
|---|---|---|---|
| TC1 | Abstract | 1 | "Human [..] release" .... . The use of human as an adjective in this sentence sounds somewhat unusual to me. Consider replacing it by "release by humans"  (also line 18) |
| TC2 | Abstract | 12 | typo: sarutarion --> saturation |
| TC3 | | 37 | "or" -> shouldn't this be "and" ? |

| | | | |
|---|---|---|---|
| TC4 | 1 | 42 | Listing the NO2I together with CLIM and MAP suggests that it is a separate instrument from the CO2I, that is confusing considering the description later (see also comment line 83) |
| TC5 | 2 | 83 | CO2I/NO2I: earlier CO2I and NO2I are described as separate. Please keep this consistent, I would advice to use CO2I/NO2I |
| TC6 | 2 | 126 | neighbored -> neighboring |
| TC7 | Fig 2 | | typo: "white colours" --> white colour |
| TC8 | Fig 2 | | Please consider decreasing the white space between the panels to make the actual figure content larger. |
| TC9 | Table 1 caption | | The caption reads rather difficult, could you rephrase it? |
| TC10 | Fig 8 caption | | "Note that the fraction is ..." there seems to be something missing in this sentence, please correct. |

---

## Author Comment (AC1)

**Response to comments by Referee #1 of EGUsphere-2024-3857**

Dear Referee,

Thank you very much for this comprehensive review of our manuscript which helped to improve the manuscript. Please find below our point-by-point responses to your comments. The comments are printed in italics and our responses are shown in upright font.

Kind regards and on behalf of all authors,

Michael Weimer

*The paper is in general written in a clear and succinct manner and the reasoning is good to follow. The subject is highly relevant for the future CO2 mission.*

*There are however a few points, where clarifications are needed and the consistency needs to be checked. The biggest question I have is (see also SC2), if – instead of using oversampling higher than one – the sampling time can be decreased? Has this been investigated? Now the investigation is reduced to integration times of 271, 117 and 66ms. What would for example 250 ms sampling time produce as a result? Then the ground sample would be about 1.8 x 1.8 km, so the ground resolution would be higher (and produce symmetric ground pixels). What are the reasons to use 271 ms? With a back of the envelope calculation (using the results from Table 1) I estimate that 208 ms integration time are needed that the 24% saturated pixels for the OSF=1 case are within the FWC limit. Are there limitations on datavolume, internal datarate, synchronization of read-outs, co-registration or other reasons to use 308ms sampling time? Is there an estimate of the level of saturation versus integration time? How large are the signal independent contributions (dark current, thermal background, offset)? Can they be reduced? Please address these points in the article.*

Authors' response: The timing of the CO2I/NO2I instrument is driven by the requirements for a) spatial sampling ($<= 4$ km$^2$) and b) SNR of the spectral radiance measurement. The ACT spatial sampling distance is given by the design of the slit homogenizer (as well as detector size and optical magnification), and is fixed to 1.8 km. This allows for a maximum of ~2.2km spatial sampling distance (SSD) in ALT. At the CO2M orbit this amounts to a sampling time of ~308 ms. We added this explanation to the manuscript in Sect. 2. Since the SWIR detectors are operated in "Integration Then Read" (ITR) readout mode, there is a readout-time of 37 ms at the end of each frame (or scanline acquisition) during these 308 ms. We added this explanation also in the manuscript before Eq. 2: "It has been decided to operate CO2I's SWIR detectors in "Integration Then Read" mode, in order to minimize bias effects from the detector's read-out electronics. In this mode, the signal of every acquired frame has to be completely read out before a new acquisition is started. During the read-out time $t_{RO}$ of about 37 ms, the signal integration is paused."

The largest possible integration time (which is reached in OSF1) is therefore 308 ms – 37 ms = 271 ms. So 271 ms is the effective integration time in mode OSF1, 308 ms - (2*37 ms)= 2*117 ms = 234 ms for OSF2, and so on for higher OSFs. A reduction of the sampling time (common to all bands) would indeed reduce the fraction of saturated pixels. However, in that case the most saturated band would dictate the maximum sampling (and hence integration) time for all other bands, as they have to be synchronized to comply with spatial co-registration. If the

SWIR1 is then taken as the driving band, the lower integration time (e.g. 208 ms) would lead to an integration time of 208 ms – 37 ms = 171 ms. This would lead to a reduction of integration time of ~37%, which would lead to lower SNR of sqrt(171 ms / 271 ms) = 21 % globally for all ground pixels. The SNR requirement is specified for rather dark reference radiances, and such reduction of SNR performance would certainly lead to non-compliance to the specification. To clarify this, we added a sentence to line 118 of the preprint: "The value of 308 ms was used to meet all demands within the mission requirements."

There are limitations on all of the mentioned parameters. However, as explained above, the 308 ms are the result of Space Segment Requirements (the instrument design, incl. data rate is then tuned to meet the specification). As pointed out above, the integration time is fixed and follows (for each OSF) from requirements and design considerations (readout mode). Signal independent contributors to the signal (and SNR), such as dark current and thermal background) can theoretically be reduced, most notably by operating detectors and optics at colder temperatures. However, these operational parameters have been fixed by the payload prime and are not within the scope of this paper. We agree that one result of this study is that also the sampling time could be reduced to avoid the caveats of an additional read-out, but this requires further studies in the future. Therefore, we added a corresponding paragraph to the conclusions (line 374): "This study used a fixed sampling period of 308 ms to meet the mission requirements. In principle, this value can be adjusted as well in order to get the optimum between SNR and saturation avoidance. Further investigations in the future could include the reduction of the sampling period instead of increasing the OSF. However, a smaller sampling period would further reduce the SNR and synchronization of the CO2I/NO2I spectrometers might become more difficult with a smaller sampling period."

***SC1 (line 86)****: CO2I -> CO2I/NO2I: the slit is also shared with the VIS spectrometer, so CO2I alone is not accurate*

Authors' response: We replaced all occurrences of CO2I by "CO2I/NO2I" where it relates to the whole instrument and not to the CO2I spectrometers only.

***SC2 (line 92)****: As an alternative to the OSF couldn't the sampling time of 308ms be reduced? With 250 ms the ground sample would be about 1.8 km x 1.8km. Why not? What are the limitations (detector limitations, datavolume,...)? Is there an estimation how short the exposure time would have to be to avoid saturation everywhere? See also separate discussion above.*

Authors' response: see our response to the main comment above. There are limitations and requirements that lead to the value of 308 ms.

***SC3 (line 110)****: The spatial sampling is not affected, but I would expect an impact on the the spatial energy distribution function. Can you please comment on this?*

Authors' response: We did idealized tests about this in the context of this study where we convolved the signal with the IFOV, assuming a perfect instrument with rectangular FOV and same sensitivity at the edge as in the center. When doing this, the signal for OSF = 2 is reduced to a minimum of 68 % when the signal **only** comes from the gap created by the detector read-out, see Figure AR1 in this response. Hence, even if the whole signal comes from a region which is never measured by the middle ALT

detector pixels, the signal still is 68 % because of the IFOV of 0.4 km. This is an extreme case and more realistically, the differences in and outside the gap will be much smaller, see e.g. right panel of this figure, where the radiance is 80 % of that in the gap and the fraction is 84 % of the radiance after 271 ms. Therefore, yes, it has some impact on the distribution of the signal but it is considered to be small, which resulted in the sentence in the manuscript.

[Figure]

[Figure]

**Figure AR1:** Normalized radiance (blue) and accumulated signal for an OSF of two (orange) for two cases: Radiance only comes from the gap for read-out (left) and the radiance is 80 % in the surrounding in comparison to that in the gap (right).

*SC4 (line 114): Only the detectors for SWIR 1 and 2 are mentioned here. What about NIR? What is the FWC of the NIR detector? Is it the same?*

Authors' response: The detector used in the VIS and NIR bands is the CIS-120 from Te2v. It has a FWC of 61.000e-, and OSF adaptation to avoid saturation is indeed planned. E.g. OSF is tunable between 3-7 in VIS and between 1-3 in NIR. However, since the SNR in these two bands is significantly less affected by multiple readouts due to the lower readout noise per acquisition (60e- vs 150e- in SWIR), it is assumed that the SNR performance in NIR will be better than in SWIR, and therefore has no significant impact on the three-band retrieval performance. We added the FWC of the NIR detector of about 61 ke- to the manuscript: "The detectors used for the NIR detector (Teledyne-E2V) and the two SWIR spectrometers of CO2I/NO2I (Lynred NGP) feature a FWC of approximately 61.000 and 650.000 electrons, respectively."

*SC5 (line 118): "a radiance spectrum with saturated pixels has to be discarded" The reasoning to discard the entire spectrum is not described clearly. I would advice to change the order of this paragraph somewhat and mention firstly (as described in line 268) that not single pixels but at least 60 are affected (what fraction is that of the spectrum?) and secondly that this impacts the straylight correction.*

Authors' response: We basically moved the description around line 268 up to line 118 with replacing 60 pixels by 3 %: "Such pixels do not yield meaningful measurements and tests showed that if saturation occurs it usually does not happen only at one spectral detector pixel but for more than 3 % of the pixels with the largest signal. Therefore, it can be expected that a large fraction of the continuum range of the spectrum is affected by saturation so that the measurement is not useful for the retrieval and a radiance spectrum with saturated pixels has to be discarded."
In addition, we replaced the introductory sentences around line 268 by the

following: "As discussed in Sect. 2, spectra including saturated measurements have to be discarded, which will reduce the spatial coverage on Earth."

**SC6 (line 126)**: *"neglect the effect… on neighbored spatial samples" . This is unclear: do you mean other viewing angles/swath angles/ spatial samples in the same frame where saturation occurs? Then it should be excluded as a whole anyways, as the straylight correction would be insufficient. Or do you mean the impact on following read-outs? (see next comment).*

Authors' response: It refers to the same frame and we added "…neighboring spatial samples in the swath" to the sentence. We did the analysis also with removing whole swaths instead of single spatial samples in the context of this study, resulting in a total fraction of 47 % of left data for OSF111 with similar spatial distribution as in Figure 5 in the manuscript. Apart from that, the results showed only minor changes. We decided to add this analysis as another appendix, now Appendix B, to the manuscript, which we refer to in line 276: "Because of possible stray light effects, we also did the analysis with removing whole swaths instead of single spatial samples, which can be found in Appendix B, and where a fraction of 47.3 % remains after filtering for saturation."

**SC7 (line 127)**: *Is there anything known about detector blooming or the effect of pixel saturation on the following (unsaturated) read-outs? Is there a memory effect? Or is the assumption here that only an individual frame is affected?*

Authors' response: Blooming effects (spatial-spectral signal spillover) would be another reason to avoid saturation. However, blooming is assumed to be eliminated by the anti-blooming functionality, which is integrated into the ROIC of the NGP detector. Anti-blooming avoids signal spillover to neighboring pixels even if the FWC is reached. Memory effects (temporal signal spillover) has indeed been detected in both the CIS-120 and NGP detectors (used for the NIR and SWIR bands, respectively). They are a topic of dedicated studies, some of which have also been published (see e.g. Gaucel et al. (2023): Remanence characterization of NGP detector in SWIR bands, https://doi.org/10.1117/12.2689982). The influence of saturation on signal persistence or memory effects is out of the scope of this paper. However, we added possible complications arising from such effects in the introduction, as an additional motivation for the study (around line 66 of the preprint): "In addition, saturation could affect subsequent measurements due to memory effects (Gaucel et al., 2023)." and added "and memory effects" to line 127.

**SC8 (line 168)**: *Can something be said about the off-nadir angles? Is the effect of saturation expected to be smaller there? And do you then assume the nadir spectrum for all viewing angles (spatial samples on the detector). This sentence seems to contradict the statement in line 126.*

Authors' response: We agree that this statement is confusing. Of course, we take into account the viewing geometry within the swath of the instrument. We replaced "nadir geometry" by "the nadir mode of CO2M" because we wanted to separate it from the glint mode, which was not used in this study.

**SC9 (line 242)**: *The numbers in the text are not consistent with the insets in Fig. 3. If you want to give ranges which include all bands, it should be 67 and 86% and 47 to 73 % for OSF 2. Or you can remove "in all bands" in line 243 and use the ranges 67-92% and 47-83%*

Authors' response: We replaced "89" by "92" % and removed "in all bands" as suggested.

**SC10 (line 256)**: *"some high SNR values have a large noise error" Could you please add an explanation why that is the case?*

Authors' response: In general, radiances in the NIR band are less sensitive to changes of $CO_2$ in the atmosphere. Therefore, the relation between noise error and SNR is not as straight-forward as for the other bands. We added the following explanation to the sentence: "[…] because radiances in the NIR band are less sensitive to changes of $CO_2$ (only indirectly due to the dependence of the retrieved $XCO_2$ on atmospheric scattering and on the air column or surface pressure) than in the other bands."

**SC11 (line 306)**: *"not done in this analysis": was this shown elsewhere? Please add a reference.*

Authors' response: We now refer to Noël et al. (2024) in the manuscript where they separated systematic from random contributions by using high- and low-pass filters. We also rephrased the complete sentence: "Note that this value includes both systematic and random errors that were not separated in this analysis, as e.g. in Noël et al. (2024)."

**SC12 (line 312)**: *"decreased by about 20%", is this due to the saturation filtering? Please clarify this in the manuscript.*

Authors' response: Yes. We added "due to the saturation filtering" at the end of the sentence.

**SC13 (line 333)**: *"glint mode could change", change in what way? Please specify what you expect.*

Authors' response: This was not in the scope of this study. We expect different results because the signals in glint mode will be different. This will probably have to be optimized for each mode and satellite separately. We also already refer to glint in the conclusions that studies of the glint mode are needed in the future.

**TC1 (line 1)**: *"Human [..] release" …. . The use of human as an adjective in this sentence sounds somewhat unusual to me. Consider replacing it by "release by humans" (also line 18)*

Authors' response: Corrected as suggested.

**TC2 (line 12)**: *typo: sarutarion --> saturation*

Authors' response: Corrected.

**TC3 (line 37)**: *"or" -> shouldn't this be "and" ?*

Authors' response: Yes, corrected.

**TC4 (line 42)**: *Listing the NO2I together with CLIM and MAP suggests that it is a separate instrument from the CO2I, that is confusing considering the description later (see also comment line 83)*

Authors' response: We added to the sentence that NO2I shares the same slit with CO2I.

**TC5 (line 83)**: *CO2I/NO2I: earlier CO2I and NO2I are described as separate. Please keep this consistent, I would advice to use CO2I/NO2I*

Authors' response: We replaced all occurrences of CO2I by "CO2I/NO2I" where the sentence relates to the whole instrument and not to the CO2I spectrometers only.

**TC6 (line 126)**: *neighbored -> neighboring*

Authors' response: Corrected.

**TC7 Fig 2**: *typo: "white colours" --> white colour*

Authors' response: Corrected.

**TC8 Fig 2**: *Please consider decreasing the white space between the panels to make the actual figure content larger.*

Authors' response: Corrected as suggested.

**TC9 Table 1 caption**: *The caption reads rather difficult, could you rephrase it?*

Authors' response: We rephrased it to: "Fraction of cloud-free spatial samples (in %) for which the radiance is between the saturation limits of OSF and OSF minus one in the 1-year dataset of simulated radiances."

**TC10 Fig 8 caption**: *"Note that the fraction is …" there seems to be something missing in this sentence, please correct.*

Authors' response: We added "to" between "related" and "the data".

---

## Author Comment (AC2)

**Response to comments by Referee #2 of EGUsphere-2024-3857**

Dear Referee,

Thank you very much for this comprehensive review of our manuscript which helped to improve the manuscript. Please find below our point-by-point responses to your comments. The comments are printed in italics and our responses are shown in upright font.

Kind regards and on behalf of all authors,

Michael Weimer

*The authors present the impact of avoiding detector saturation on the precision and sampling frequency of XCO2 retrieved from CO2M data by using the Fast atmospheric traCe gAs retrievaL (FOCAL), and possible CO2M observation scenarios based on simulation experience. I understand that this activity is important to assess and optimize the CO2M observation scenario, and maximize the available observation data from CO2M with requested precision and accuracy. However, some description and assumption are unclear or missing in the text.*

*In this study, the sampling duration time for detector is fixed at 308ms. I understand the sampling duration time is composed of the integration time of detector and the time for readout. The times for reading out the signal might be fixed due to the limitation of signal transfer speed between detector response itself and electronical chain, and it is indicated about 37ms in this study. In contrast, the integration time for detector might be able to customize within the allowable time periods which is depended on the character of detector, and this function will lead the optimization of the total performance of instrument. In other words, if the CO2M instrument will be adopted the reduced integration time (234 (=2\*117) ms) with the oversampling factor (OSF) =1, it might lead higher SNR and small foot print size than that of OSF=2 (2\*(117+37) ms) case, which is proposed by the authors.*

*Probably, some limiting conditions on detector operation such as the fixed sampling duration at 308 ms are existed but not clearly described in this manuscript. If these limitations are not existed, the other observation scenario has to be considered, and be proposed. Then, the clarification of limiting conditions on detector operation are important for this study.*

Authors' response: The timing of the CO2I/NO2I instrument is driven by the requirements for a) spatial sampling (<= 4 km2) and b) SNR of the spectral radiance measurement. The ACT spatial sampling distance is given by the design of the slit homogenizer (as well as detector size and optical magnification), and is fixed to 1.8 km. This allows for a maximum of ~2.2km spatial sampling distance (SSD) in ALT. At the CO2M orbit this amounts to a sampling time of ~308 ms. We added this explanation to the manuscript. In theory, an optimal sampling time could be derived to avoid saturation, but in practice, it would further reduce the SNR which is already close to the limit and the synchronization of the four spectrometers would be more difficult for a faster sampling time. To clarify this, we added a sentence to line 118 of the preprint: "The value of 308 ms was used to meet all demands within the mission requirements." We agree that one result of this study is that also the sampling time could be reduced to avoid the caveats of an additional read-out, but this requires further studies in the future. Therefore, we added a corresponding paragraph to the conclusions (line 374):

"This study used a fixed sampling period of 308 ms to meet the mission requirements. In principle, this value can be adjusted as well in order to get the optimum between SNR and saturation avoidance. Further investigations in the future could include the reduction of the sampling period instead of increasing the OSF. However, a smaller sampling period would further reduce the SNR and synchronization of the CO2I/NO2I spectrometers might become more difficult with a smaller sampling period."

*In addition, the authors mentioned that the operational OSF will be determined during commissioning phase of CO2M mission. I understand that this manuscript is focused on the simulation-based evaluation. However, it is also important to have a plan; how to determine the operational OSF during commissioning phase? Because, the authors mentioned that the retrieval of CO2M will be performed among 3 parties. Then, the clear procedure or plan for determination of OSF might be required.*

Authors' response: The planning of the commissioning phase for CO2M is not finalized yet, the critical design review (CDR) is currently in discussion. Part of this study's motivation is that it will contribute to the planning of the commissioning phase. We agree that this has not been outlined in the current manuscript and hence, we added the following sentence to the introduction (line 62) to clarify this: "[...] in order to contribute to the planning of the commissioning phase for CO2M." In addition, we added a final sentence to the conclusions: "These results are intended to be used for the planning of the commissioning phase for CO2M."

**Specific comments.**

**Abstract**

*Page 1, line 4: coverage. -> "wide" coverage*

Authors' response: Corrected.

*Page 1, line 8: sampling -> sampling "frequency" or sampling "number".*

Authors' response: We changed it to "spatial coverage" instead of sampling.

**4. Defining scenarios avoiding detector saturation**

*Page 8, line 173: What are the criteria for acceptance of constant OSF settings all over the globe? To optimize the parameters, the acceptance levels both global coverage, retrieval precision and accuracy are important. The authors should add the explanation.*

Authors' response: The criteria for the satellite mission are defined in the Mission Requirement Document (MRD, ESA, 2020), but they assume the OSF scenario 111. In addition, providing acceptance levels for the coverage (e.g. something like "it should be larger than 95 %") is misleading because sources of greenhouse gases are localized emissions and coverage should be ensured in the areas where CO2 and CH4 are emitted. Therefore, providing specific numbers, which we assume is meant by "acceptance levels" here, is not the right way to do it in this study. In addition, we show later in the manuscript that the retrieval precision with OSF222 is still close to the values of the VEG50 scenario used in the MRD so that the comparability is already provided in the manuscript.

**6.2 Impact on coverage**

*Page 11, line 341: coverage -> spatial coverage.*

Authors' response: We replaced both occurrences of "coverage" by "spatial coverage".

*Page 12, line 280: In the figure 5, it seems that the middle east countries are indicated as the saturation area. Regarding the Oil & Gas emissions from middle east countries, it might have some impact on emission estimates. In parallel, the authors are focusing on XCO2 on this manuscript. If the authors are considering XCH4, the scenario with OSF111 might be reconsidered. Then, the authors should add the detail explanation for these considerations.*

Authors' response: We agree with the referee's statements about the Oil & Gas emissions from the Middle East which would have to be filtered out when using OSF 111. We added the sentence "Note that important emissions from oil and gas industry on the Arabian Peninsula would have to be filtered out when using OSF 111." to the discussion around line 274. On the other hand, we disagree with the referee's statement that OSF111 might be reconsidered when thinking about XCH4 in this region because the issue with saturation is independent of the species so that basically all data would have to be filtered out also for XCH4 in this region. In contradiction to the referee's statement, this is another example that an OSF larger than 1 should be used for CO2M.

**6.3 Impact on XCO2**

*Page 16, Figure 7: The location of legend is not suited and have to move on top or bottom of figures. The current location is hard to find it.*

Authors' response: We moved the legend below the figure.

*The trends of noise error are almost similar among the OSF's (OSF111, 222, 232, and 333). I understand the most impacted parameters are the multiple number of read-out, digitization and video chain noises. Typically, figure 7(e) is clearly indicated the relationship among the OSF's. However, figure7 (g) and (i) suggests some anomalous relationship in the seasonal variation. Then, the authors should add the explanation what is the course of these anomalies. In addition, figure 7 (a) suggests that the setting of OSF222 creates the minimum noise error condition during winter season (October, November, December). In this case, how to conclude the optimization of OSF for CO2M mission? So, the authors also should add the explanation for these conditions.*

Authors' response: The anomalies come from the different number of data filtered due to saturation. As can be seen in the figure, the anomalies are within the tropical regions between 40°S and 40°N where most of the deserts on the Earth's surface are located, see also panels (h) and (j) showing the number of data in these latitude bands. As the deserts include most of the saturated spectra, filtered out during pre-processing, and they have a large signal if they are not filtered out their noise error will be smaller than the average. Therefore, the median noise error for OSF 111 is increased when filtering for saturation. We added the following sentence to line 315 of the manuscript: "As the SNR is connected to the noise error (see Fig. 4) and the signal over deserts is usually larger than the average, the median noise error increases when the saturated spectra of desert regions are filtered out, which explains the anomalies in the noise error globally and in the near-tropical latitude bands (panels a, g and i)."

**7. Summary and Conclusions**

*Page 19, line 383: How to optimize the OSF during commissioning phase? The authors conclude that the scenarios 111 and 222 seems to be favorable for CO2M in the future with the FOCAL. In this study, the conclusion is based on the simulated one-year subset of radiance data, and not considered the retrieval results from Fusional-P-UOL-FP and RemoTAP. To optimizing the operational scenarios during the commission phase of CO2M, the authors also have to consider the realistic procedure how to determine the OSF? The authors should add the explanation.*

Authors' response: It is clear that all parameters have to be considered, but the planning of the commissioning phase is not finalized yet. This study provides the planning of the commissioning phase with parameters to be considered, which is actually the motivation of this study. That is why we added clarifications to introduction and conclusions as stated in our response to your major comment above.

---

## Author Response (AR2)

Dear Referee #1,

Thank you for this second review of our manuscript. We agree that it was not clear from the previous version of the manuscript why 271 ms was chosen as integration time for this study. Therefore, we added both suggested figures to the manuscript with their description in the appendix which we hope are helpful in understanding these issues. Please find our point-by-point responses below.

Best regards and on behalf of all authors,

Michael Weimer
* * *
Thank you for the detailed explanations in your answers. The additional analysis you have performed on discarding the entire swath if a part is saturated is very valuable. This gives a more realistic estimate of the remaining usable data. Do you also have a percentage for the data available after postprocessing for this case? It might be worthwhile adding this number if you have it readily available.

Authors' Response: We added the suggested panels to the figure and added a short explanation to the sentence describing the figure in the main text.

In your rework you have addressed many of my points in a satisfactory way. However, on the sampling interval/integration time discussion I still miss some fundamental information:
While acknowledging that there are constraints from mission requirements and hardware, it is also important for a scientific study to show very clearly what the fundamental limitations are in such a situation. This analysis is needed to make sure that the best choices and tradeoffs between the different requirements can be made for this mission. A possible non-compliance on the spatial sampling might be acceptable if the SNR and coverage could be improved significantly.

I do not fully follow the argumentation in your answer, of course the signal decreases if a shorter integration time is used, but when comparing the case for OSF=2, an exposure time of 234ms surely has a higher SNR than 2* 117ms? The total integration time is the same, but with two reads there will be twice the read-out noise and also a gap in the spatial sampling due to the read-out time.
So please add to your manuscript (without consideration for the prescribed values for the integration times) a table or curve with the integration time vs % of saturated pixels. What would be the optimal integration time for the CO2/NO2 instrument to avoid saturation? For SWIR 1 24.51% (value from Table 2) of the pixels are saturated for 271 ms, when would it be 10%, 5%, 1%? And what SNR would be expected then for NIR/SWIR1/SWIR2?

Authors' Response: We thank the referee for providing more detail in their concerns about the integration time issue. We calculated the A and B parameters for all sampling periods between 0 and 400 ms. We added the figure and the details of the description to the manuscript as another appendix. In summary, the fraction of global spatial samples that are affected by saturation increases from zero to about 40 % between sampling periods of 150 to 400 ms and OSF1. For larger OSF values, this fraction is smaller, as expected. The expected average SNR increases

with larger sampling periods, therefore a sampling time is used that is a trade-off between small fraction of saturation and largest SNR.